# SYMBOLIC EQUATION SOLVING VIA REINFORCEMENT LEARNING

## ABSTRACT

Machine-learning methods are gradually being adopted in a great variety of social, economic, and scientific contexts, yet they are notorious for struggling with exact mathematics. A typical example is computer algebra, which includes tasks like simplifying mathematical terms, calculating formal derivatives, or finding exact solutions of algebraic equations. Traditional software packages for these purposes are commonly based on a huge database of rules for how a specific operation (e.g., differentiation) transforms a certain term (e.g., sine function) into another one (e.g., cosine function). Thus far, these rules have usually needed to be discovered and subsequently programmed by humans. Focusing on the paradigmatic example of solving linear equations in symbolic form, we demonstrate how the process of finding elementary transformation rules and step-by-step solutions can be automated using reinforcement learning with deep neural networks.

## 1 INTRODUCTION

Deep learning and artificial intelligence have become ubiquitous tools in industry and academia, demonstrating automatic data-processing and control capabilities with unprecedented speed and accuracy. In general, deep-learning schemes are known to excel in statistical and approximation settings, where many similar problems and/or solutions exist. However, machine-learning models often struggle in discrete settings that require exact solutions and precise sequences of actions, such as symbolic mathematics (Choi (2021); Ornes (2020); Bubeck et al. (2023); Wolfram (2023b)). Consider, for instance, an equation like $3x + 2c = x + 4$ with the unknown $x$ and a symbolic parameter $c$. It has one and only one exact solution, $x = 2 - c$ (up to symmetries), but no "approximately acceptable" answers, and a number of equivalence transformations in the right order needs to be carried out to find this solution.

Traditional software packages to deal with exact mathematics are so-called computer algebra systems (CAS) such as Mathematica, Maple, Matlab, or SymPy. These programs can analyze and manipulate mathematical expressions in symbolic form, i.e., in an analytically exact manner. With their functionality covering almost all areas of mathematics, they have become an indispensable tool in scientific research and education. Common use cases include term simplification, calculus, and solving algebraic equations.

The core of such CAS is typically a database of rules that encode how a specific term pattern $A$ is transformed into another pattern $B$ if operation $T$ is applied,

$$\text{operation } T : \quad \text{term } A \; \rightarrow \; \text{term } B \,. \tag{1}$$

To implement a differentiation module, for example, one elementary rule could be "$\partial/\partial x : \sin x \rightarrow \cos x$." A powerful CAS requires a vast number of such elementary transformation rules. All of these rules first had to be found by humans, a process that has lasted for millennia in the form of mathematical research. In a second step, humans must implement the discovered rules as computer programs. Evidently, this process could benefit greatly from techniques that enable computers to start from a certain set of definitions and established rules and combine them to discover and implement new transformation rules on their own. Moreover, finding viable approaches to do so will eventually help to make machine-learning models more adept at mathematics and problems requiring exact solutions in general.

Here we develop a deep reinforcement-learning scheme that can autonomously find elementary transformation rules for a key component of CAS functionality, the symbolic solution of algebraic equations. To illustrate this task, two basic rules of such a CAS module could be

$$\text{solve for } x: \ x + a_0 = b_0 \ (a_0, b_0 \in \mathbb{C}) \ \rightarrow \ x = b_0 - a_0 \tag{2}$$

and

$$\text{solve for } x: a_1 x = b_0 \ (a_1 \in \mathbb{C} \setminus \{0\}, b_0 \in \mathbb{C}) \ \rightarrow \ x = b_0/a_1 \,. \tag{3}$$

To solve the more complicated problem $a_1 x + a_0 = b_0$, we can define a third rule that decomposes the problem: To solve $f(x) + a_0 = b_0$ for $x$, first solve for $f(x)$ (using rule (2)), then solve for $x$ (using rule (3) in this case). This way, complex equations can be disentangled and eventually solved.

Our goal is to obtain a model that proposes such step-by-step strategies for solving linear equations: It should analyze the structure of the equation at every step and devise an expedient transformation towards the final solution.

## 2 SETUP

### 2.1 DEEP REINFORCEMENT LEARNING

We set up a reinforcement learning (RL) framework that facilitates the automatic discovery and implementation of transformation rules. As usual, it involves an *agent*, whose role is to suggest the transformation rules, and an *environment*, which consists of the equation to be solved and some auxiliary information. We briefly recall a few key concepts here to introduce notation and refer to the reviews (Arulkumaran et al. (2017); Sutton & Barto (2018); Marquardt (2020)) for a more in-depth exposition as well as to Appendix A for technical details of our implementation.

The RL agent learns through a feedback loop: In every time step $t$, it makes an observation $s_t = v(q_t)$ of the state $q_t$ of the environment and subsequently chooses an action $a_t$ according to a conditional probability distribution ("policy") $\pi(a_t \mid s_t)$. The environment responds to this action by changing its state into a new one, $q_{t+1}$. Furthermore, it issues a reward $r_t$, which is positive if the action was beneficial for the task, negative if it was adverse, or zero if the implications are unclear. On the basis of these rewards, the agent may update its policy in an attempt to maximize the expected future return $V_\pi(s) := \mathbb{E}_\pi[\sum_{t=0}^\infty \gamma^t r_t \mid s_0 = s]$ for all states $s$, where $\gamma \in [0, 1)$ is a discount factor to ensure convergence.

We adopt the $Q$-learning approach based on the quality function

$$Q(s, a) := \mathbb{E}\left[\sum_{t=0}^\infty \gamma^t r_t \mid s_0 = s, a_0 = a\right]. \tag{4}$$

It is obtained from $V_\pi$ by splitting off and conditioning on the first action $a$, and subsequently following a greedy policy with respect to $Q$ itself, meaning that we always choose the action that maximizes $Q$ in the present state, $a_t = \arg\max_a Q(s_t, a)$, for $t \geq 1$. The $Q$ function of the optimal policy, which yields the largest discounted return, satisfies the Bellman equation (Arulkumaran et al. (2017); Sutton & Barto (2018); Marquardt (2020))

$$Q(s_t, a_t) = \mathbb{E}\left[r_t + \gamma \max_a Q(s_{t+1}, a)\right]. \tag{5}$$

Implementing double deep $Q$-learning (Hasselt et al. (2016)), we train a neural network $f_\theta$ with parameters $\theta$ to represent $Q$. The network receives the observed state $s$ and maps it onto a vector whose components are the values of $Q$ for each action $a$, i.e., $Q(s, a) = [f_\theta(s)]_a$. The parameters $\theta$ are updated with the goal of solving the Bellman equation (5), see Appendix A.3 for details.

### 2.2 EQUATION CLASSES

As explained above, we focus on the task of finding exact solutions for linear equations. More specifically, we consider two main types of equation classes. The first one consists of linear equations of the general form

$$a_0 + a_1 x = a_2 + a_3 x \,, \tag{6}$$

where $x$ is the variable to solve for and the coefficients $a_i$ are numbers taken from $\mathbb{Z}$, $\mathbb{Q}$, $\mathbb{R}$, or $\mathbb{C}$; the specific choices constitute several subclasses.

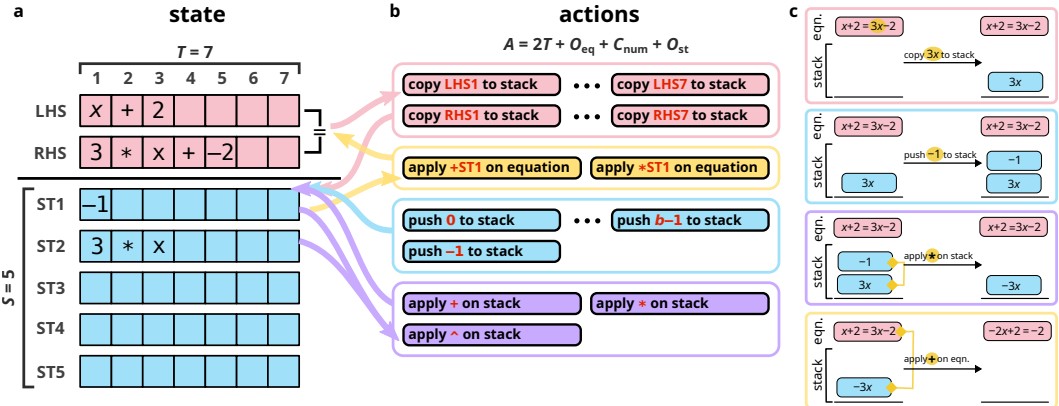

Figure 1: Main components of the reinforcement-learning framework for symbolic equation solving. (a) The state comprises the left- and right-hand sides of the equation as well as a stack of up to $S$ additional terms. Every term consists of (up to) $T$ elementary units (i.e., operators, variables, numbers, parentheses). (b) The agent can choose from four classes of actions: copy (sub)terms from the equation to the stack ($2T$ actions); perform an equivalence transformation of the equation ($O_{\mathrm{eq}}$); push a predefined numerical constant to the stack ($C_{\mathrm{num}}$); or apply a mathematical operation on the stack ($O_{\mathrm{st}}$). (c) Examples of how the state changes under each of the four classes of actions.

The second type involves one additional symbolic parameter $c$ and consists of equations of the form

$$a_0 + b_0\,c + (a_1 + b_1\,c)\,x = a_2 + b_2\,c + (a_3 + b_3\,c)\,x\,,\tag{7}$$

where $a_i, b_i \in \mathbb{Z}, \mathbb{Q}, \mathbb{R}$, or $\mathbb{C}$ as before. Importantly, mastering this problem enables the machine to solve systems of (linear) equations as well by solving one equation for the first variable (e.g., $x$), substituting the solution into the second equation, and solving for the second variable ($c$), and so on if there are more than two variables.

During training, we generate an equation of the form (6) or (7) whenever a new task is assigned by sampling the coefficients $a_i$, $b_i$ randomly from an appropriate distribution (see Appendix A.1 for details). For testing, we use fixed sets of example equations generated in the same way and provided as supplementary material for reference. We also explore an adversarial approach where training problems are generated by a second RL agent. As explained in more detail below, the goal of this generator agent is to find as simple a problem as possible such that the solver agent fails.

## 2.3 Environment and actions

Let us now introduce our RL framework for solving algebraic equations. The key ingredients are illustrated in Fig. 1. The first component of our RL environment is the equation, initially given in the form (6) or (7), for example. It consists of two mathematical expressions (terms), namely the left-hand side (LHS) and the right-hand side (RHS), cf. Fig. 1a.

The agent is to carry out a sequence of equivalence transformations that either cast the equation into the form

$$x = (\text{something independent of } x)\tag{8}$$

or eliminate the variable $x$ from it if the original equation was ill-defined. Once this is achieved, we consider the problem to be solved. If the agent fails to reach such a state within $t_{\mathrm{max}}$ elementary steps, we consider the attempt to be unsuccessful.

To suggest the right sequence of transformations, the agent must be able to perform all the mathematical operations occurring in the equation or, more precisely, the corresponding inverse operations. We implement this ability by giving the agent access to a "stack calculator" that can manipulate terms in symbolic form. This is the second component of our RL environment (cf. Fig. 1a). It consists of a stack of up to $S$ terms. Whenever a new term is added to the stack, it is inserted in the top position and all entries move down accordingly. We can compose arbitrary mathematical operations by taking a fixed number of terms from the top of the stack, feeding them into a predefined function,

and pushing the result back onto the stack. For example, the operation "+" removes the first two terms from the stack, adds them together, and deposits the sum on the stack again. Provided that the stack calculator supports all relevant operations, this enables the agent to prepare and eventually execute all necessary equivalence transformations to solve the equation.

Specifically, the agent can choose from four different types of actions for this purpose, cf. Fig. 1b. The first type allows it to copy a (sub)term from the left- or right-hand side of the equation to the stack (cf. Fig. 1c for an example). To this end, every term is decomposed into up to $T$ elementary units, which are operators (+, $\star$, ^), variables ($x$, $c$), numbers ($2$, $-\frac{1}{3}$, $\sqrt{2}$), or parentheses. The agent can select any of those elementary units for copying. For units consisting of operators or parentheses, the entire associated subterm will be cloned. This gives a total of $2T$ "copy" actions.

The second action type involves pushing one of $C_{\mathrm{num}}$ predefined numerical constants to the stack. These typically include the numbers $0$ and $1$ (basis of the binary number system), $-1$ (frequently needed for inverse operations), and $\mathrm{i}$ (imaginary unit) in the case of complex-valued coefficients $a_i$, $b_i$ in Eqs. (6)–(7). Pushing several $0$s and/or $1$s sequentially is interpreted as providing the binary digits of an integer.

The third action type applies one of $O_{\mathrm{st}}$ specific mathematical operations on the stack as outlined above. In our case, $O_{\mathrm{st}} = 3$ with addition "+", multiplication "$\star$" and exponentiation "^".

Finally, the fourth type facilitates the actual equivalence transformation by performing one of $O_{\mathrm{eq}}$ specific mathematical operations on both sides of the equation, possibly involving additional arguments from the stack. In our case, in particular, we have $O_{\mathrm{eq}} = 2$ with addition "+" and multiplication "$\star$", both of which take the top entry from the stack as the second operand.

By skillful assembly of these actions in the right order, the agent can then transform any equation of the initial forms (6)–(7) into the goal state (8). The four steps taken in Fig. 1c provide one example of a sequence that serves to eliminate the variable $x$ from the right-hand side. A full, machine-solved example can be found in Fig. 2d.

To avoid any human bias about what the optimal strategy is, we only issue a positive reward $r_{\mathrm{slv}}$ when the agent reaches the goal and achieves a state of the form (8) or eliminates $x$ for ill-defined problems. Furthermore, we deduct from this final reward $r_{\mathrm{slv}}$ a penalty $p_{\mathrm{st}}$ for every term that is left on the stack and a penalty $p_{\mathrm{as}}$ for any assumptions that are necessary during the transformation (e.g., $x \neq 0$ if the equation is divided by $x$), see also Appendix A.6. The idea of these penalties is to encourage concise solution strategies that avoid redundant and unnecessary steps.

This general framework can be extended to other types of algebraic equations by suitably adjusting the available operations for equation and stack manipulations. Furthermore, our philosophy is for the agent to focus on the equation-solving logic: The agent should understand the structure of the equation and devise the correct inverse transformation. However, it should not be bothered with the algebraic simplification of mathematical expressions (e.g., $3 \times 2 \rightarrow 6$ or $-2x + 5x \rightarrow 3x$). Such simplifications are instead performed automatically in the environment using SymPy (Meurer et al. (2017)), cf. Appendix A.7. In a future, more autonomous and machine-learned CAS, we envision that this process is handled by a different RL agent or module that is specialized in term simplification. Likewise, other SymPy features could be replaced or added using machine learning.

## 3 RESULTS

### 3.1 PURELY NUMERICAL COEFFICIENTS

We concentrate on equations of type (6) with real-valued coefficients first. In Fig. 2a, we monitor the test success rate (i.e., the fraction of successfully solved problems from a fixed set of test equations) as a function of the training time $\tau$ (i.e., the number of parameter updates). Training initially uses only integer coefficients ($a_i \in \mathbb{Z}$ in Eq. (6)), corresponding to the solid lines in the figure. After an initial exploration period of around $5 \times 10^6$ training epochs with only sporadic success, the performance drastically improves, reaching success rates of almost $100\%$ on the test set with integer coefficients (top-left panel of Fig. 2a). At the same time, we observe that only about $50\%$ of equations with rational coefficients are solved successfully (top-right panel of Fig. 2a). Hence we start a second training scheme using equations with coefficients $a_i \in \mathbb{Q}$ from this point (dashed

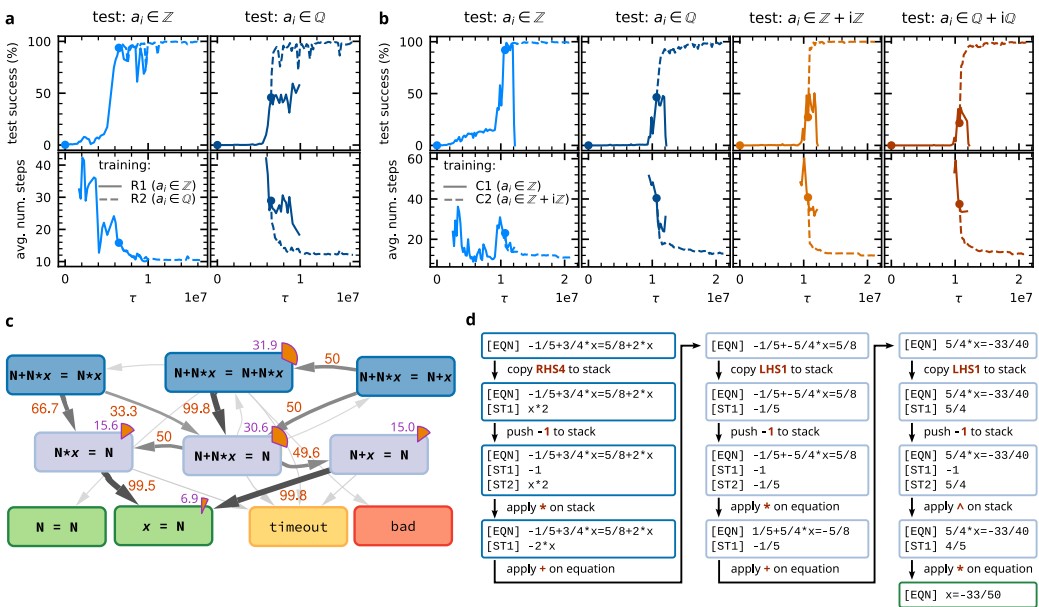

Figure 2: Test success and solution strategies for equations of type (6) with (a) real-valued coefficients or (b–d) complex-valued coefficients. (a, b) Test success and average number of elementary steps to success vs. training time $\tau$ for various test datasets (see column heading) and maximum number of steps $t_{max} = 100$. Average number of steps (bottom row) only shown if test success is $\geq 2\,\%$. (a) Solid (R1): training with integer $a_i$; dashed (R2): training with rational $a_i$. (b) Solid (C1): training with real-integer $a_i$; dashed (C2): training with complex-integer $a_i$. (c) Summary graph for the solution strategy of (C2) after $\tau = 1.955\times10^7$ training epochs, analyzed for the complex-rational test dataset. States are collected into superstates according to the form of the equation, disregarding left-right symmetry and the stack. Disk sectors and numbers (in purple) indicate the relative number of elementary steps spent in the respective superstates (in $\%$, only shown if $\geq 1\,\%$). Transitions between superstates are indicated by arrows, with their relative frequencies compared to all transitions from a fixed state given by the numbers in red (in $\%$, only shown if $\geq 1\,\%$). Special states: *timeout:* SymPy processing of the selected action exceeds $10\,$s; *bad:* state cannot be represented as a neural-network tensor (e.g., term size exceeds $T$, numerical overflow). (d) Example sequence of elementary transformations suggested by the network from (c) to solve $-\frac{1}{5} + \frac{3}{4}x = \frac{5}{8} + 2x$.

lines in Fig. 2). This way, we eventually achieve success rates $\geq 99.8\,\%$ on both the integer- and rational-coefficient test sets. We point out that there is no conceptual generalization in admitting irrational coefficients because the state encoding presented to the agent only provides floating-point representations of numerical constants (cf. Appendix A.2), i.e., the agent will deal with rational and general real-valued coefficients in the same way.

In the lower panels of Fig. 2a, we also show the average number of steps required for solved tasks (restricted to machines with success rates $\geq 2\,\%$). Initially, solutions are found somewhat accidentally and therefore require a large number of steps. Once the agent has managed to achieve some consistency, the required number of steps goes down and gradually decreases further as the strategy is fine-tuned.

For complex-valued coefficients $a_i$ in Eq. (6), the results in Fig. 2b show very similar characteristics. Starting with training on real-integer coefficients (solid lines), the agent makes gradual progress until the success rate abruptly improves at around $\tau = 10^7$ (top-left panel of Fig. 2b). At this point, the agent also starts managing to solve some of the problems with more general coefficient types (second to fourth columns in Fig. 2b). However, continued training with real-integer coefficients soon becomes unstable, leading to diverging parameter updates, a common side effect of deep $Q$-learning (Sutton & Barto (2018); van Hasselt et al. (2018); Wang & Ueda (2022)). Switching to complex-integer coefficients instead (dashed lines) yields further improvement of the test success, with eventual success rates $\geq 98.9\,\%$ across all different test sets. We particularly highlight that

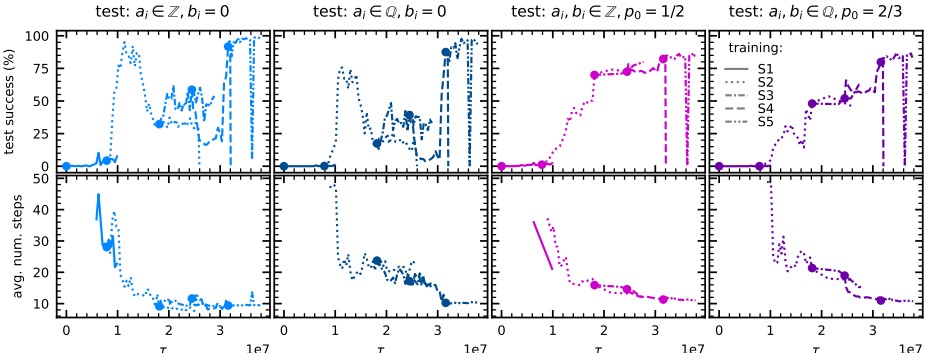

Figure 3: Test success and average number of steps for equations of type (7) with mixed numerical (real-valued) and symbolic coefficients. Test types: see column headings; maximum number of steps: $t_{\max} = 100$. Solid (S1): training with $a_i, b_i \in \mathbb{Z}$ and $a_0 = b_0 = a_3 = b_3 = 0$ or $a = 1 = b_1 = a_2 = b_2 = 0$, $p_0 = 0$, learning rate $\eta = 0.05$; dotted (S2): training with $a_i, b_i \in \mathbb{Z}$, $p_0 = 1/2$, $\eta = 0.05$; dash-dotted (S3): training with $a_i, b_i \in \mathbb{Z}$, $p_0 = 2/3$, $\eta = 0.01$; dashed (S4): training with $a_i, b_i \in \mathbb{Q}$, $p_0 = 2/3$, $\eta = 0.05$; dash-dot-dotted (S5): training with $a_i, b_i \in \mathbb{Q}$, $p_0 = 2/3$, $\eta = 0.01$. Average number of steps for successful transformations (bottom row) only shown if test success is $\geq 2\,\%$.

the agent generalizes from training on complex-integer coefficients to near-perfect success rates on complex-rational coefficients (cf. top-right panel of Fig. 2b).

In Fig. 2c, we illustrate the solution strategy of one of the best agents on the equations with complex-rational coefficients. For clarity, we group states according to their equation structure. The graph visualizes the transitions between the resulting superstates, highlighting their relative frequencies among all transitions from a given superstate. Typically, an initial sequence of transformation steps serves to eliminate the variable from one side of the equation (top to middle layer in Fig. 2c), which most commonly results in equations of the form $a_0 + a_1 x = b_0$ (middle node of the middle layer). From here, the agent proceeds in two different ways without significant preference: Either it subtracts $a_0$, leading to the N*x=N pattern (left node of the middle layer) or it divides by $a_1$ and ends up in the N+x=N superstate (right node). This is one incidence of a more general observation that the strategies discovered by the agent tend to be more fine-grained than the recipes typically taught to humans. The final transformation then consists of the pertinent division or subtraction, leading to the solution state x=N. A concrete example with the detailed states and elementary transformations as suggested by this agent is given in Fig. 2d.

## 3.2 MIXED SYMBOLIC AND NUMERICAL COEFFICIENTS

Next we turn to the second type of equations, cf. Eq. (7), which feature a second symbolic variable $c$ besides the unknown $x$. Note that this problem is considerably more complex than type (6) because of much greater term variety and sizes, which additionally lead to a larger action space. Test results for various training schemes and test types are shown in Fig. 3, monitoring the success rate and the average number of steps required for successfully solved equations.

As before, we follow a training strategy that gradually increases the problem complexity. We begin with a simplified version where either $a_0 = b_0 = a_3 = b_3 = 0$ or $a_1 = b_1 = a_2 = b_2 = 0$ in Eq. (7) (label S1, solid lines). Once the agent has learned to deal with these structurally simpler equations, we switch to the full form, focusing on $a_i, b_i \in \mathbb{Z}$ first (S2 and S3, dotted and dash-dotted lines). The S2 and S3 training schemes differ in the frequency of symbolic coefficients and some of the hyperparameters (see Appendix A.8). During the S2 stage, the agent first learns to solve equations with purely numerical coefficients, similar to type (6), after around $\tau \approx 10^7$ epochs (cf. the first and second columns of Fig. 3), but initially struggles with equations involving the parameter $c$ (third and fourth columns). As it improves its solution capabilities for the latter type of symbolic problems, it temporarily "forgets" how to deal with purely numerical coefficients between epochs $\tau \approx (1.5 \ldots 3) \times 10^7$. Hence we raise the frequency of equations with purely numerical coefficients

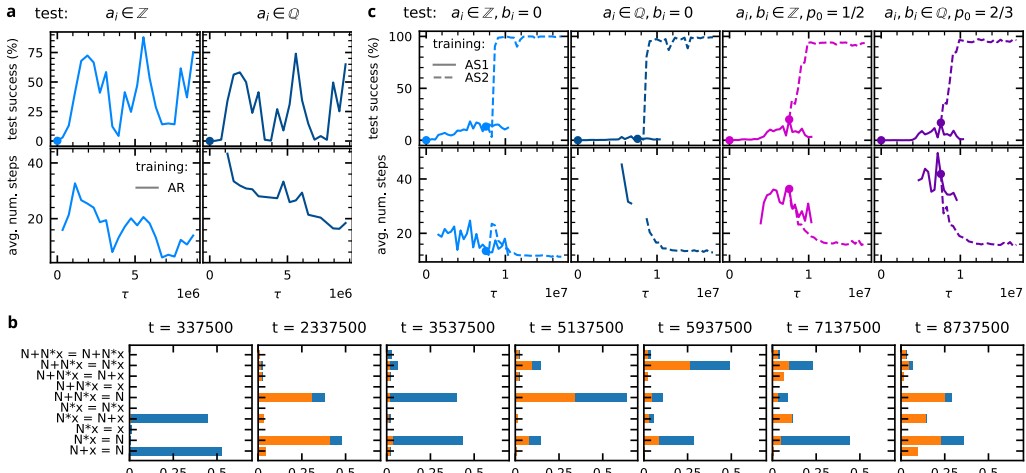

Figure 4: Characteristics of the adversarial reinforcement-learning approach. (a) Test success and average number of steps for equations with purely numerical coefficients (test types: see column headings). Generator initialization: $x = a \in \mathbb{Q}$ (see also Appendix A.1). (b) Histograms of the equation patterns produced by the generator (blue) and the fraction of successfully recovered solutions by the solver (orange). (c) Test success and average number of steps for equations with mixed symbolic and numerical coefficients (test types: see column headings). Solid (AS1): adversarial training, generator initialization $x = (a_1 + b_1 c)/(a_2 + b_2 c)$ with $a_i, b_i \in \mathbb{Q}$, $p_0 = 1/2$; dashed (AS2): fixed-distribution training with equation type (7), $a_i, b_i \in \mathbb{Q}$, $p_0 = 2/3$. Learning rates and greediness are adapted according to online estimates of the learning progress, see Appendix A.5 for details. Average number of steps for successful transformations (bottom row in (a) and (c)) only shown if test success is $\geq 2\,\%$.

in the training data by increasing the probability $p_0$ of vanishing $b_i$ (S3), which leads to a modest improvement on the purely numerical test sets again.

More consistent solutions for all types of test datasets are only achieved after changing to schemes with $a_i, b_i \in \mathbb{Q}$ (S4 and S5, dashed and dash-dot-dotted lines). In this way, we eventually obtain simultaneous success rates $\geq 86.5\,\%$ on equations with both purely numerical and mixed numerical/symbolic coefficients, meaning that the agent has essentially learned to deal with the problem types (6) and (7).

## 3.3 ADVERSARIAL TASK GENERATION

So far, we generated training tasks from fixed distributions and gradually increased their complexity manually when the agent appeared to have mastered a class of problems. For equations involving symbolic coefficients, in particular, this required several adaptations of the training task distribution.

To streamline this process and to achieve an even more autonomous discovery of solution strategies, we introduce an adversarial learning scheme in the following. We train a second RL agent, the *generator*, simultaneously with the *solver* agent. In every episode, the generator creates a problem task, which is then passed to the solver. The generator operates in a similar environment as the solver (cf. Fig. 1), consisting of an equation (LHS and RHS) and a stack. Unlike the solver, however, the generator starts from a solution, i.e., an equation of the form (8), and tries to manipulate this equation such that it becomes unsolvable for the solver. To this end, the generator has access to the same set of actions plus an additional "submit" action to pass its current equation to the solver. After submission, the generator receives a positive reward $r_{\text{fool}}$ if the solver fails to recover the solution, and no reward otherwise. In addition, a small penalty $p_{\text{step}}$ is deducted for every step of the generator; see also Appendix A.6. This way, the generator is encouraged to find the simplest possible problem that overpowers the solver. As the solver's capabilities improve, the generator will have to increase the problem complexity to keep pace.

Focusing on equations with purely numerical coefficients again first, we show the test performance of a single adversarially trained solver agent in Fig. 4a, utilizing the same sets of test equations as in Fig. 2a. Comparing these two sets of results, we observe that the adversarially trained solver achieves decent success rates faster than the agent learning from a fixed training distribution, although it ultimately fails to reach the same level of reliability (maximal success rate $\geq 74\%$ across both test sets). In Fig. 4b, we illustrate how the generator adapts its output as the solver becomes more successful: The blue bars show the relative frequency of classes of generated equations, whereas the orange bars correspond to the solver's success on these equations. While those classes are only a coarse indicator of the equation difficulty, it is clearly visible how the diversity and complexity of the generator increases during training.

Finally, we turn to equations with mixed symbolic and numerical coefficients again, for which we show test performance results in Fig. 4c. Beginning with an adversarial learning scheme (AS1, solid lines), we observe some progress in the test success rates. Yet the eventually achieved success is not overly impressive, partly because the generator fails to explore the full spectrum of equations in the test dataset. However, it turns out that the adversarially trained solver agent (AS1) forms an excellent basis for fixed-distribution training as in the previous subsection. Switching to such a scheme with equations involving symbolic and real-valued numerical coefficients after $t \approx 7.5 \times 10^6$ episodes (AS2, dashed lines), we eventually achieve success rates $\geq 94.2\%$ across all test datasets. Hence this agent outperforms the best agents found under the manual scheme (without adversarial pretraining, cf. Fig. 3) and learns to solve equations of all test types reliably. Moreover, no further adaptation of the training distribution or hyperparameters is necessary and the total training time is reduced considerably with adversarial pretraining.

## 4 Discussion

### 4.1 Related studies

Machine-learning techniques have been adopted to tackle a variety of mathematical problems and tasks (He (2023)), from assisting with proofs and formulating conjectures (Kaliszyk et al. (2018); Bansal et al. (2019); Davies et al. (2021)) to numerical solutions of algebraic (Liao & Li (2022)) and differential (Han et al. (2018)) equations to efficient algorithms for high-dimensional arithmetic (Fawzi et al. (2022)).

Among the more closely related applications, a particular example that also involves processing terms in a mathematically exact manner is symbolic regression. Here the goal is to deduce an analytical formula that describes a given dataset, using a predefined set of available operations. Specific examples using machine learning include retrieving physical laws from measurements (Udrescu & Tegmark (2020)) and finding recurrence relations from example sequences (d'Ascoli et al. (2022)).

A previously studied task which often involves aspects of equation solving are math-text exercises, i.e., formulations of mathematical problems in natural language. Besides the actual arithmetic, the challenge here is to extract the abstract math problem from the verbal description. This type of problem has been tackled most successfully with language models (Wei et al. (2022); Drori et al. (2022); Lewkowycz et al. (2022); Bubeck et al. (2023); Wolfram (2023a)), i.e., neural networks specialized in natural language processing. The models are trained on a huge corpus of human-authored texts and, when primed for math and science reasoning, exemplary problem sets and solutions. Their solution strategies thus imitate human approaches: The neural network generates a textual output which it estimates to be the most likely solution for the given question based on the training corpus, with remarkable albeit far from perfect success rates. By construction, this method is unlikely to succeed when applied to previously unexplored domains where human-solved examples are unavailable. Another caveat is that language models will always produce an answer and sell it formally convincingly, but there is actually no guarantee for correctness of the answer itself nor of the underlying reasoning. A partial remedy for this problem is to "outsource" the actual logic, for instance by translating the problem into a query to specialized software such as a CAS (Wolfram (2023a)). In this case, however, the actual solution strategy must have been known and algorithmically implemented a priori by humans. By contrast, our present approach does not presume human knowledge about solution strategies, with the effect that it may fail to produce a solution, but it will never generate a wrong answer.

As another example, Poesia et al. (2021) introduced a new reinforcement-learning scheme dubbed "contrastive policy learning" and subsequently applied it to find solutions to linear equations with integer coefficients. Besides the much smaller domain, the first major difference in their approach is that the agent does not need to discover inverse transformations by itself, but finds them among the available actions at each step. In our case, the agent must instead construct inverse operations through the stack, with the advantage of requiring fewer basic actions, but at the expense of additional elementary steps. The second major difference is that the focus of Poesia et al. (2021) is more on term simplification using basic algebraic axioms, a task that we have deliberately outsourced to SymPy (see last paragraph of Sec. 2.3). Hence both the motivation and the actual learning task are quite different. Indeed, one of the main conclusions of Poesia et al. (2021) is that deep $Q$-learning—as we adopt it here—fails to find reliable solution strategies within their framework.

Yet another important example of standard CAS functionality, symbolic integration, was tackled by machine-learning methods by Lample & Charton (2020). Here the task is to find an indefinite integral of a given mathematical expression. Their approach uses supervised learning of so-called sequence-to-sequence models, which are commonly adopted for natural language processing. Similarly to the math-text problems (see above), and contrary to the philosophy of our approach, the machine does not discover the underlying elementary transformation rules on its own, but is trained on a large dataset of exemplary function–integral pairs to predict an integral (output) from a function (input). The model achieves impressive success rates, though some reservations regarding the test set and generality apply (Davis (2019); Ornes (2020)). Furthermore, the machine's "reasoning" behind successful (or failed) integration attempts remains intransparent to humans.

## 4.2 Conclusions and outlook

Our work explores a new way to adopt machine learning for exact mathematics, a domain that has proved to be troublesome for traditional approaches. It can be seen as a first step towards general machine-learning strategies to discover laws of mathematical reasoning and deduction. Specifically, we devised a neural-network-powered RL agent that acquires the capability of solving linear equations by composing elementary equivalence transformations. It builds on established rules for handling mathematical terms and discovers autonomously how such manipulations can be adopted to master a new domain. In some sense, this situation is similar to a middle-school student who already knows the rules of basic algebra, but still needs to learn about the concept of equations and how to solve them. More generally, mathematical research can be seen as a gradual exploration of how axioms and previous insights can be combined to reveal something nontrivial that was not immediately apparent from the original components. Here we showcase a strategy for how machines can be utilized for this endeavor. Our demonstration uses the arguably simple example of solving linear equations. However, the proposed scheme based on a symbolic stack calculator can be directly extended to other types of algebraic equations by a suitable adaptation of the available operations, and we expect that the idea of having an RL agent explore a suitable "theory space" can prove itself effective in other computer-algebra domains as well and aid mathematical research in general.

A primary design principle of our approach was to provide an unbiased setting in which the agent explores how to solve the task without reference to what humans may perceive as the best strategy. Hence we deliberately avoided certain techniques that could have accelerated the learning process, but introduce a human bias, such as supervised (pre)training, intermediate rewards, or preselected inverse transformations. Crucially, once the machine finds a successful recipe, its strategy remains comprehensible to humans thanks to the composition from simple elementary transformations, such that humans can subsequently learn it from the machine.

It will merit further study to extend the solution capabilities to other types of equations and inequalities. Furthermore, it will be of both fundamental and practical interest to develop similar schemes for other types of computer algebra problems, e.g., term simplification, symbolic integration, limits and series representations, and eventually tackle tasks that are not amenable using presently available software.

## Acknowledgments

[hidden during double-blind peer review]

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

# A  IMPLEMENTATION DETAILS

## A.1  TASK SAMPLING

In the first part of this work (Secs. 3.1 and 3.2), sample problems during training are generated by sampling the coefficients $a_i, b_i$ in Eqs. (6)–(7) randomly and independently from fixed probability distributions. Let $U_n^{\mathbb{N}}$ denote the uniform distribution on $\{1, 2, \ldots, n\}$ and $U_n^{\mathbb{Z}}$ the uniform distribution on $\{-n, -n+1, \ldots, n\}$. We then have: for $a_i \in \mathbb{Z}$, $a_i \sim U_{10}^{\mathbb{Z}}$; for $a_i \in \mathbb{Q}$, $a_i = p/q$ with $p \sim U_{50}^{\mathbb{Z}}$, $q \sim U_{10}^{\mathbb{N}}$; for $a_i \in K + iK$, $a_i = \alpha_{\mathrm{re}} + i\alpha_{\mathrm{im}}$ with $\alpha_{\mathrm{re}}$ and $\alpha_{\mathrm{im}}$ independent and randomly drawn from $K = \mathbb{Z}$ or $\mathbb{Q}$ according to the aforementioned schemes. The numbers $b_i$ multiplying the symbolic constant $c$ in type (7) are set to zero with probability $p_0$ and otherwise sampled from the same distribution as the $a_i$. The fixed test datasets (cf. Appendix B) are generated from the same distributions.

In the second part of this work (Sec. 3.3), training tasks are generated by an additional RL agent that adapts to the capabilities of the solver as outlined in Sec. 3.3. The solutions from which the generators start are drawn randomly as $x = a_1$ (AR, Fig. 4a,b) or $x = (a_1 + b_1 c)/(a_2 + b_2 c)$ (AS1, Fig. 4c) with $a_i, b_i \in \mathbb{Q}$ distributed as before.

## A.2  STATE ENCODING

To feed the observed state $s$ into the neural network $f_\theta(s)$ for the $Q$ function, we need to translate $s$ into a suitable representation. We recall that the state consists of $S + 2$ terms (stack, LHS, RHS), each of which can contain up to $T$ elementary units, cf. Fig. 1a. We encode each of these terms in a feature plane as illustrated in Fig. 5, which is a $(C + N) \times T$ matrix. Its $T$ columns correspond to the term's elementary units (in standard infix notation, see also below), and the rows encode the contents of those units. The first $C$ rows contain binary indicators for whether or not the unit is of a specific type. There are $O_{\mathrm{st}}$ rows for the supported operators (here: +, *, ^), two rows for the left and right parentheses, one row for the variable $x$, and one row for each additional symbolic constant. Furthermore, there is one row to indicate whether the unit contains a constant (numerical or symbolic). Finally, there are $N$ additional rows to encode the floating-point values of the numerical constants ($N = 1$ for real-valued coefficients, $N = 2$ for possibly complex-valued coefficients). Hence the overall input to the neural network is an $(S + 2) \times (C + N) \times T$-dimensional tensor, and its output is an $A$-dimensional tensor where $A = 2T + O_{\mathrm{eq}} + C_{\mathrm{num}} + O_{\mathrm{st}}$ is the total number of available actions (cf. Fig. 1b). In our examples, the input and output dimensions range from 280 to 1071 and 18 to 45, respectively, cf. Tables 3 and 4 below.

The state representation passed on to the neural network is based on the standard infix notation of mathematical expressions, meaning that a term whose top-level operator takes multiple arguments (such as +, *) is expressed by interleaving the arguments with copies of the operator symbol ("$x + c + 1$"). A more natural encoding of mathematical terms is an expression tree with the nodes

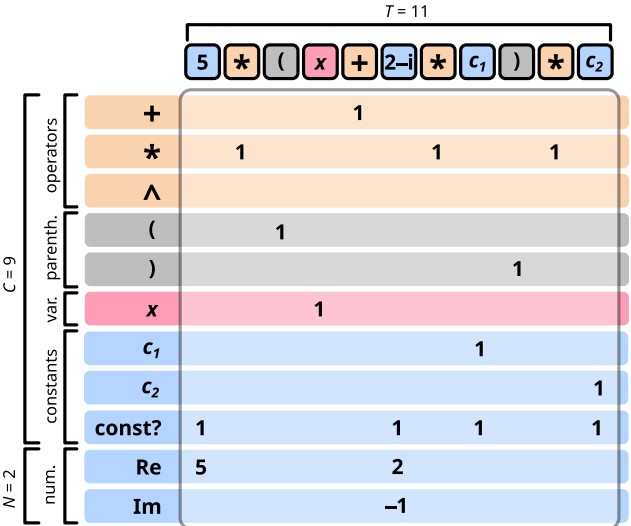

Figure 5: Illustration of the state-tensor representation using feature planes for the example term $5x(x + (2 - \mathrm{i})c_1)c_2$. Every term is mapped to a $(C + N) \times T$ tensor, whose columns correspond to the elementary units (operators, parentheses, variables, numbers) of the expression. The $C$ character rows hold binary indicators of whether a specific (type of) symbol occurs in the respective position of the term. The $N$ number rows encode the floating-point values of numerical constants. Empty entries correspond to the value zero. A full state representation consists of $S + 2$ such term tensors (left- and right-hand sides of the equation as well as the stack entries).

corresponding to the operators and the leaves representing variables and numbers, and this is also the format of our internal (SymPy-based) representation. It clearly maps out the hierarchy and priorities of the operations and avoids the need for parentheses and/or implicit operator precedence rules, but does not have a natural embedding into the layout of network tensors. An alternative to the infix representation is a prefix notation, which avoids operator precedence rules and parentheses, provided that the number of arguments is fixed for each operator. However, the latter condition introduces an artificial evaluation hierarchy, and the prefix notation can lead to very nonlocal representations in the sense that arguments appear far away from their operators. Altogether, we could not identify a clear practical advantage of either prefix or infix notation and thus stuck with the latter, which has the additional advantage of familiarity and thus better interpretability by humans.

For the floating-point representation of (the real and imaginary parts of) numerical constants in the $N$ number rows of the state tensor (cf. Fig. 5), we rescale and cap them to obtain typical values of order unity for numerical stability. Concretely, for (the real or imaginary part of) a coefficient $a$, the corresponding entry of the state tensor is set to $a/100$ if $|a| \leq 500$ and otherwise causes a numerical overflow.

### A.3 NETWORK ARCHITECTURE AND TRAINING ALGORITHM

We adopt neural networks $f_\theta$ with three (four) hidden layers of $8000, 4000, 2000$ ($16\,000, 8000, 4000, 2000$) units for the problems of type (6) (type (7)). We use the standard ReLU activation function ($\sigma(x) = \max\{0, x\}$) as our nonlinearity for all layers except the last one, which is linear.

We update the network parameters $\theta_\tau$ according to the double deep $Q$-learning scheme (Hasselt et al. (2016)) with experience replay (Mnih et al. (2015)): In every epoch $\tau$, the agent explores $p$ steps in the current environment following an $\varepsilon$-greedy policy with respect to its present $Q$-function estimate $Q(s, a) = [f_{\theta_\tau}(s)]_a$, meaning that the highest-valued action is chosen with probability $1 - \varepsilon$ and a random action otherwise. Invalid actions are masked (cf. Appendix A.4) and the (non)greediness $\varepsilon$ is gradually decreased with the training progress (cf. Appendix A.5). The observed 4-tuples $(s_t, a_t, r_t, s_{t+1})$ of explored transitions are stored in a replay memory with maximum capacity $M = 5 \times 10^5$. (Note that $t$ denotes the time within the current episode of the RL

environment, i.e., the number of actions taken by the agent since the current task was assigned. It should not be confused with the training time $\tau$, which counts the total number of parameter updates/training epochs.)

Thereafter, we sample a batch of $B$ transitions $\omega := (s, a, r, s')$ uniformly at random from the replay memory and calculate the current estimate of $Q(s, a)$ (LHS of the Bellman equation (5)) for all state-action pairs $(s, a)$ in the batch by evaluating the network function $f_{\theta_\tau}$. To obtain a sufficiently decorrelated estimate of the RHS of the Bellman equation (5), its $Q$ values are estimated from a second *target network*, whose parameters $\hat{\theta}_\tau$ are aligned with the parameters $\theta_\tau$ of the *online network* only every $\hat{\tau}$ epochs. The squared residual of Eq. (5) associated with the transition $\omega$ is thus

$$\ell_\theta(\omega) := \left([f_\theta(s)]_a - r - \gamma[f_{\hat{\theta}}(s')]_{a'}\right)^2, \qquad (9)$$

where $a' := \arg\max_a[f_\theta(s')]_a$ and, by convention, $f_{\hat{\theta}}(s') \equiv 0$ if $s'$ is a terminal state (solved or invalid). Our loss function is the mean over those residuals for all transitions in the batch, $L_\theta := \frac{1}{B}\sum_\omega \ell_\theta(\omega)$. The online-network parameters $\theta_\tau$ are then updated using gradient descent with learning rate $\eta$, $\theta_{\tau+1} := \theta_\tau - \eta\nabla_\theta L_{\theta_\tau}$.

Our actual implementation relies on the PyTorch library (Paszke et al. (2019)). The relevant hyper-parameters and their values in our experiments are summarized in Tables 1–4 below.

### A.4 INVALID ACTION MASKING

For training and inference, actions that are invalid in a given state are masked, i.e., they are excluded from the available options. Concretely, this means that the greedy policy chooses the action with the highest $Q(s_t, a) = [f_\theta(s_t)]_a$ among the valid actions $a$ in the current state $s_t$, and the random policy uniformly selects one of these valid actions. Invalid actions are:

- "copy `LHS`$n$ (`RHS`$n$) to stack" if the term size of LHS (RHS) is less than $n$. For example, the term `2+4*x` has five elementary units, so any copy actions for positions beyond 5 are masked;

- "apply `f` on stack" if `f` requires $n$ arguments, but the number of stack entries is less than $n$;

- "apply `f` on equation" if `f` requires $n$ arguments from the stack, but the number of stack entries is less than $n$;

- multiply the equation by zero;

- "apply `^` on stack" with arguments $a$ and $b$ (i.e. $a^b$) if $a = 0$ or $b \notin \mathbb{Z}\setminus\{0\}$.

Note that the last type of action is not mathematically invalid, but can lead to ambiguities (e.g., regarding the sign of a variable when $b = \frac{1}{2}$) that would require special care. Since transformations of this kind are not needed within the domain of equations under study, we exclude them to avoid such subtleties.

For the generator agent within the adversarial learning approach, the "submit" action is masked if "LHS = RHS" is not a linear equation.

### A.5 GREEDINESS AND LEARNING-RATE SCHEDULES

During training, we dynamically adjust the (non)greediness of our $\varepsilon$-greedy policy with the training progress. We use two different schemes: (i) exponentially decaying schedules of the general form

$$\varepsilon(\tau) = (\varepsilon_i - \varepsilon_f)\,e^{-\tau/T_\varepsilon} + \varepsilon_f, \qquad (10)$$

which interpolate between the initial value $\varepsilon_i$ ($\tau = 0$) and asymptotic final value $\varepsilon_f$ ($\tau \to \infty$), where $\tau$ is the training time (number of parameter updates); (ii) adaptive schedules of the form

$$\varepsilon(s) = (\varepsilon_i - \varepsilon_f)(1 - s)^{\alpha_\varepsilon} + \varepsilon_f, \qquad (11)$$

where $s \in [0, 1]$ is an online estimate of the current agent capabilities and $\varepsilon_i$ ($\varepsilon_f$) is the maximum (minimum) value. For solver agents, $s$ is the fraction of successfully solved equations within the last 100 episodes. For generator agents, $s$ is the minimum of the success rate to generate valid equations

and the failure rate of the solver within the last 100 episodes. The values of $\varepsilon_i$, $\varepsilon_f$, $T_\varepsilon$, and $\alpha_\varepsilon$ in our experiments are listed in Tables 3 and 4.

The learning rates are either kept fixed for a specific training configuration,

$$\eta(\tau) = \eta = \text{const}, \tag{12}$$

or adjusted dynamically similarly as in (11),

$$\eta(s) = (\eta_i - \eta_f)(1 - s)^{\alpha_\eta} + \eta_f. \tag{13}$$

Here $s$ is the same online quality estimate as before. The adopted values in the experiments are listed in Tables 3 and 4, too.

### A.6 REWARDS

*Solver agents.* As explained in Sec. 2.3, the total reward when an equation is solved is given by the success reward $r_{\text{slv}}$, reduced by penalties for remaining stack entries ($p_{\text{st}}$) and assumptions about unknowns ($p_{\text{as}}$). Our explicit formula to calculate the final reward is

$$r = r_{\text{slv}} - \frac{n_{\text{st}}}{S}p_{\text{st}} - n_{\text{as}}p_{\text{as}}, \tag{14}$$

where $n_{\text{st}}$ is the number of stack entries in the final state, $S$ the stack capacity, and $n_{\text{as}}$ the number of assumptions. Assumptions are added whenever an operation involves a variable ($x$ or $c$) and is not generally valid for all real or complex values of this variable. For example, when the action "apply ^ on stack" is selected with base $c + 1$ and exponent $-1$, the assumption "$c + 1 \neq 0$" is added. As another example, when choosing the action "apply $\star$ on equation" with the stack argument $2x - 3$, the assumption "$2x - 3 \neq 0$" is added. If the same assumption is required twice, it will only be added once. Note that assumptions cannot generally be avoided if the equation involves variables besides the unknown to solve for, but the agent should be encouraged to find transformations that require as few of them as possible.

To discourage pointless manipulations and auxiliary calculations, we furthermore consider the possibility of issuing a negative reward $r_{\text{so}}$ in case of a "stack overflow," i.e., when the agent chooses to push a term to the stack even though the latter already holds $S$ terms. (The action is still carried out, with the result that the lowest entry on the stack, which had been there the longest, is discarded.)

*Generator agents.* For generator agents as introduced in Sec. 3.3, the total reward after the "submit" action is given by

$$r = \begin{cases} 0 & \text{if solver found solution} \\ r_{\text{fool}} & \text{otherwise} \end{cases} - \frac{n_{\text{st}}}{S}p_{\text{st}} - n_{\text{as}}p_{\text{as}}. \tag{15}$$

Here the stack and assumption penalties are handled like for the solver agents.

After all other actions, a negative reward (step penalty) is issued to encourage fast submission:

$$r = -p_{\text{step}}. \tag{16}$$

The concrete values for rewards and penalties in the numerical experiments are listed in Tables 3 and 4.

### A.7 TERM PROCESSING

As mentioned in the main text, our RL environment automatically simplifies mathematical expressions when updating its state. This includes the following steps in order:

- if constants can be complex-valued numbers or symbolic, call SymPy's `expand` function;
- if constants can be symbolic, call SymPy's `cancel` function. Without doing so, spurious dependencies on the variable can sometimes remain and the environment can fail to detect that an equation has been solved. E.g., the simplification $[-f(x)][f(x)]^{-1} \to -1$ may fail;
- call SymPy's `collect` function to collect terms in the variable to be solved for, e.g., $3x + 2cx - 1 \to (3 + 2c)x - 1$;

- group (and simplify if possible) numerical arguments in commutative (sub)terms, e.g., $1 + x + \frac{3}{2} + 2c + \sqrt{2} \to (\frac{5}{2} + \sqrt{2}) + x + 2c$;
- randomly shuffle the order of operands in commutative (sub)terms. This improves training stability and generalization. It also entails that the environment is nondeterministic.

We emphasize that we do not use SymPy's heuristic `simplify` function as it is computationally quite expensive and the results can vary depending on term structure, variable names, etc.

## A.8 HYPERPARAMETERS

Hyperparameters characterizing the RL environment and the training algorithm are summarized in Tables 1 and 2, respectively. The concrete values adopted in the numerical experiments are listed in Tables 3 and 4. We did not specifically optimize the environment hyperparameters (Table 1). For the training hyperparameters (Table 2), we usually explored several different training schemes, mostly focusing on the learning rate $\eta$, the greediness $\varepsilon$, the update frequencies $p$ and $\hat{\tau}$ of the online and target networks, and, to a lesser extent, the discount rate $\gamma$ and the batch size $B$.

We did not perform a systematic grid search to find globally optimal parameters due to the large number of hyperparameters and the inherent randomness of the training process, which may entail significant performance differences between training runs with the same parameters due to the instability of deep $Q$-learning (Sutton & Barto (2018); van Hasselt et al. (2018); Wang & Ueda (2022)). However, a more quantitative account of the dependence on the various hyperparameters is provided in Tables 5–16, where we show success rates for variations of the models from Figs. 2–4 (see also Tables 3 and 4). Note that the success rates in Tables 5–13 (non-adversarial framework) are based on the $\varepsilon$-greedy policy and smaller, random test sets of 500 equations drawn from the same distributions as used for training. Hence the results can vary slightly compared to the analysis shown in Figs. 2–3, which used a greedy policy ($\varepsilon = 0$) and fixed sets of test equations. Furthermore, the quoted best success rates might be surpassed during further training with the respective hyperparameters.

Table 1: Constants characterizing the reinforcement-learning environment.

| Symbol | Name | Description |
|---|---|---|
| $A$ | action count | number of actions available to the RL agent, output dim. of the neural network |
| $C$ | character rows | number of rows to indicate the type of elementary term units |
| $C_{\mathrm{num}}$ | constants count | number of numerical constants that can be pushed directly onto the stack |
| $N$ | number rows | number of rows to indicate values of numerical constants |
| $O_{\mathrm{eq}}$ | equation operations | number of operations that can be performed on the equation |
| $O_{\mathrm{st}}$ | stack operations | number of operations that can be performed on the stack |
| $S$ | stack size | max. number of terms on the stack |
| $T$ | term size | max. number of elementary units of any term |

Table 2: Hyperparameters of the neural network architecture and deep $Q$-learning algorithm.

| Symbol | Name | Description |
|---|---|---|
| $H$ | hidden layers | number of hidden layers |
| $M$ | replay capacity | number of $(s_t, a_t, r_t, s_{t+1})$ transitions in the replay memory |
| $p$ | online update frequency | number of exploration steps per parameter update of the online network |
| $\hat{\tau}$ | target update frequency | number of online parameter updates per update of the target network |
| $\hat{\varepsilon}$ | target update greediness | $\hat{\theta} \leftarrow (1 - \hat{\varepsilon})\hat{\theta} + \hat{\varepsilon}\theta$, default: $\hat{\varepsilon} = 1$ |
| $\gamma$ | discount rate | see Eq. (4) of the main text |
| $\varepsilon$ | greediness | probability for random action selection during training |
| $\varepsilon_\mathrm{i}$ | maximal greediness | see Eqs. (10) and (11) |
| $\varepsilon_\mathrm{f}$ | minimal greediness | see Eqs. (10) and (11) |
| $T_\varepsilon$ | greediness decay time | see Eq. (10) |
| $\alpha_\varepsilon$ | greediness exponent | see Eq. (11) |
| $\eta$ | learning rate | see below Eq. (9) of the main text and Eqs. (12) and (13) |
| $\eta_\mathrm{i}$ | maximal learning rate | see Eq. (13) |
| $\eta_\mathrm{f}$ | minimal learning rate | see Eq. (13) |
| $\alpha_\eta$ | learning rate exponent | see Eq. (13) |
| $\mu$ | momentum | default: $\mu = 0$ |
| $B$ | batch size | number of replay transitions used for gradient estimates |
| $t_\mathrm{max}$ | maximum steps | maximum number of elementary steps (actions) for each problem before aborting the episode |
| $r_\mathrm{slv}$ | success reward | maximal reward upon solving an equation, see Eq. (14) |
| $r_\mathrm{fool}$ | fooling reward | maximal generator reward when fooling the solver, see Eq. (15) |
| $r_\mathrm{so}$ | stack overflow penalty | (negative) reward for pushing terms on the stack beyond its capacity |
| $p_\mathrm{st}$ | stack penalty | penalty for remaining stack entries on success, see Eqs. (14) and (15) |
| $p_\mathrm{as}$ | stack penalty | penalty for assumptions on success, see Eq. (14) and (15) |

Table 3: Parameter values of the solver networks in the numerical experiments, cf. Figs. 2–4 of the main text

| Symbol | Values | | | | | | | | | | | |
|---|---|---|---|---|---|---|---|---|---|---|---|---|
| | R1 | R2 | C1 | C2 | S1 | S2 | S3 | S4 | S5 | AR | AS1 | AS2 |
| $A$ (output dim.) | 18 | | 19 | | | | 42 | | | 20 | 44 | |
| $C$ | 7 | | 7 | | | | 8 | | | 7 | 8 | |
| $C_{\mathrm{num}}$ | 3 | | 4 | | | | 3 | | | 4 | 4 | |
| $N$ | 1 | | 2 | | | | 1 | | | 2 | 2 | |
| $O_{\mathrm{eq}}$ | 2 | | 2 | | | | 2 | | | 3 | 2 | |
| $O_{\mathrm{st}}$ | 3 | | 3 | | | | 3 | | | 3 | 3 | |
| $S$ | 5 | | 5 | | | | 5 | | | 5 | 4 | |
| $T$ | 5 | | 5 | | | | 17 | | | 5 | 17 | |
| input dim. | 280 | | 350 | | | | 1071 | | | 315 | 1020 | |
| total parameters | 42 290 018 | | 42 852 019 | | | | 185 250 042 | | | 44 555 020 | 184 438 044 | |
| $H$ | 3 | | 3 | | | | 4 | | | 4 | 4 | |
| $M$ | $5\times10^5$ | | $5\times10^5$ | | | | $5\times10^5$ | | | $5\times10^5$ | $5\times10^5$ | |
| $p$ | 4 | | 4 | | 4 | 4 | 8 | 4 | 4 | 8 | 8 | |
| $\hat{\tau}$ | 100 | | 100 | | | | 100 | | | 100 | 100 | |
| $\gamma$ | 0.9 | | 0.9 | | | | 0.9 | | | 0.9 | 0.9 | 0.95 |
| $\varepsilon$ | Eq. (10) | | Eq. (10) | | | | Eq. (10) | | | Eq. (11) | Eq. (11) | |
| $\varepsilon_{\mathrm{i}}$ | 1 | 0.3 | 1 | 0.3 | 1 | 0.3 | 0.6 | 0.2 | 0.2 | 0.5 | 0.5 | 0.3 |
| $\varepsilon_{\mathrm{f}}$ | 0.1 | 0.05 | 0.1 | 0.1 | 0.1 | 0.1 | 0.1 | 0.05 | 0.01 | 0.1 | 0.1 | |
| $T_{\varepsilon}$ | 5M | | 5M | | | | 5M | | | 5M | 5M | |
| $\alpha_{\varepsilon}$ | — | | — | | | | — | | | 1 | 1 | |
| $\eta$ | 0.05 | 0.01 | 0.05 | 0.01 | 0.05 | 0.05 | 0.01 | 0.05 | 0.01 | Eq. (13) | Eq. (13) | |
| $\eta_{\mathrm{i}}$ | — | | — | | | | — | | | 0.05 | 0.05 | |
| $\eta_{\mathrm{f}}$ | — | | — | | | | — | | | 0.005 | 0.005 | |
| $\alpha_{\eta}$ | — | | — | | | | — | | | 0.5 | 0.5 | |
| $B$ | 128 | | 128 | | | | 128 | | | 128 | 128 | |
| $t_{\mathrm{max}}$ | 100 | | 100 | | | | 100 | | | 100 | 100 | |
| $r_{\mathrm{slv}}$ | 3 | | 3 | | | | 3 | | | 3 | 3 | |
| $r_{\mathrm{so}}$ | −0.25 | | −0.25 | | | | −0.25 | | | 0 | 0 | |
| $p_{\mathrm{st}}$ | 1 | | 1 | | | | 1 | | | 1 | 1 | |
| $p_{\mathrm{as}}$ | 0.25 | | 0.25 | | | | 0.25 | | | 0.25 | 0.25 | |
| test dataset | 1000 equations | | 1000 equations | | | | 10 000 equations | | | 10 000 equations | 10 000 equations | |

Table 4: Parameter values of the generator networks in the numerical experiments, cf. Fig. 4 of the main text

| Symbol | Values | |
|---|---|---|
| | AR | AS1 |
| $A$ (output dim.) | 21 | 45 |
| $C$ | 7 | 8 |
| $C_{\mathrm{num}}$ | 4 | 4 |
| $N$ | 2 | 2 |
| $O_{\mathrm{eq}}$ | 3 | 2 |
| $O_{\mathrm{st}}$ | 3 | 3 |
| $S$ | 5 | 4 |
| $T$ | 5 | 17 |
| input dim. | 315 | 1020 |
| total parameters | 44 556 021 | 184 440 045 |
| $H$ | 4 | 4 |
| $M$ | $5 \times 10^5$ | $5 \times 10^5$ |
| $p$ | 8 | 8 |
| $\hat{\tau}$ | 100 | 100 |
| $\gamma$ | 0.9 | 0.9 |
| $\varepsilon$ | Eq. (11) | Eq. (11) |
| $\varepsilon_{\mathrm{i}}$ | 0.5 | 0.5 |
| $\varepsilon_{\mathrm{f}}$ | 0.1 | 0.1 |
| $T_{\varepsilon}$ | — | — |
| $\alpha_{\varepsilon}$ | 1 | 1 |
| $\eta$ | Eq. (13) | Eq. (13) |
| $\eta_{\mathrm{i}}$ | 0.01 | 0.01 |
| $\eta_{\mathrm{f}}$ | 0.001 | 0.001 |
| $\alpha_{\eta}$ | 0.5 | 0.5 |
| $B$ | 128 | 128 |
| $t_{\mathrm{max}}$ | 100 | 100 |
| $r_{\mathrm{fool}}$ | 3 | 3 |
| $r_{\mathrm{so}}$ | 0 | 0 |
| $p_{\mathrm{st}}$ | 1 | 1 |
| $p_{\mathrm{as}}$ | 0.25 | 0.25 |

Table 5: Best test success rates on random test sets of 500 equations of type (6) with $a_i \in \mathbb{Z}$ (cf. A.1) for hyperparameter variations of model R1 (cf. Fig. 3 of the main text and Table 3). First row is the reference (R1), subsequent rows show variations; hyperparameter values are only shown if they differ from the reference.

| | hidden layers | $M$ | $p$ | $\hat{\tau}$ | $\hat{\varepsilon}$ | $\gamma$ | $T_\varepsilon$ | $\eta$ | $\mu$ | $B$ | success rate |
|---|---|---|---|---|---|---|---|---|---|---|---|
| **R1 (ref.)** | $8000, 4000, 2000$ | $5\times10^5$ | 4 | 100 | 1 | 0.9 | $5M$ | 0.05 | 0 | 128 | 91 % |
| | | | | | | | $10M$ | | | | 78 % |
| | $4000, 2000, 1000$ | $1\times10^4$ | 1 | | | | $10M$ | | | | 10 % |
| | $4000, 2000, 1000$ | $5\times10^4$ | 1 | | | | $10M$ | | | | 77 % |
| | $4000, 2000, 1000$ | $2\times10^5$ | 1 | | | | $10M$ | | | | 80 % |
| | $4000, 2000, 1000$ | $1\times10^5$ | 1 | | | | $10M$ | | | 32 | 15 % |
| | $4000, 2000, 1000$ | $1\times10^5$ | 1 | | | | | | | | 42 % |
| | $4000, 2000, 1000$ | $1\times10^5$ | 1 | | | | $10M$ | | | | 93 % |
| | $4000, 2000, 1000$ | $1\times10^5$ | 1 | | | | $10M$ | 0.02 | | 512 | 15 % |
| | $4000, 2000, 1000$ | $1\times10^5$ | 1 | | | | $20M$ | | | | 38 % |
| | $4000, 2000, 1000$ | $1\times10^5$ | 1 | | | | $20M$ | 0.02 | | 512 | 15 % |
| | $4000, 2000, 1000$ | $1\times10^5$ | 1 | | | | $10M$ | | | 512 | 10 % |
| | $4000, 2000, 1000$ | $1\times10^5$ | 1 | 10 | 0.1 | | $10M$ | | | | 20 % |
| | $4000, 2000, 1000$ | $1\times10^5$ | 1 | 10 | 0.01 | | $10M$ | | | | 85 % |
| | $4000, 2000, 1000$ | $1\times10^5$ | 1 | 10 | 0.01 | | $10M$ | | 0.1 | | 54 % |
| | $4000, 2000, 1000$ | $1\times10^5$ | 1 | 1000 | | | $10M$ | | | | 10 % |
| | $4000, 2000, 1000$ | $1\times10^5$ | 1 | | | 0.95 | $10M$ | | | | 15 % |
| | $4000, 2000, 1000$ | $1\times10^5$ | 2 | | | | $10M$ | | | | 49 % |
| | $4000, 2000, 1000$ | $1\times10^5$ | 8 | | | | $10M$ | | | | 15 % |
| | $4000, 2000, 1000$ | $1\times10^5$ | 1 | | | | $10M$ | 0.01 | 0.1 | | 15 % |
| | $4000, 2000, 1000$ | $1\times10^5$ | 1 | | | | $20M$ | | 0.1 | | 5 % |
| | $4000, 2000, 1000$ | $1\times10^5$ | 1 | | | | $10M$ | | 0.1 | | 64 % |
| | $4000, 2000, 1000$ | $1\times10^5$ | 1 | | | | $10M$ | | 0.2 | | 83 % |

Table 6: Best test success rates on random test sets of 500 equations of type (6) with $a_i \in \mathbb{Z} + i\mathbb{Z}$ (cf. A.1) for hyperparameter variations of model R2 (cf. Fig. 3 of the main text and Table 3). First row is the reference (R2), subsequent rows show variations; hyperparameter values are only shown if they differ from the reference.

| | Hyperparameters $\hat{\varepsilon}$ | success rate |
|---|---|---|
| **R2 (ref.)** | 1 | **97** % |
| | 0.1 | 71 % |

Table 7: Best test success rates on random test sets of 500 equations of type (6) with $a_i \in \mathbb{Z}$ (cf. A.1) for hyperparameter variations of model C1 (cf. Fig. 3 of the main text and Table 3). First row is the reference (C1), subsequent rows show variations; hyperparameter values are only shown if they differ from the reference.

| | Hyperparameters | | | | success rate |
|---|---|---|---|---|---|
| | $p$ | $T_\varepsilon$ | $\eta$ | $\mu$ | |
| **C1 (ref.)** | 4 | $5M$ | 0.05 | 0 | **90** % |
| | 1 | | | | 88 % |
| | | $10M$ | | | 88 % |
| | | | 0.01 | 0.2 | 8 % |

Table 8: Best test success rates on random test sets of $500$ equations of type (6) with $a_i \in \mathbb{Z} + \mathrm{i}\mathbb{Z}$ (cf. A.1) for hyperparameter variations of model C2 (cf. Fig. 3 of the main text and Table 3). First row is the reference (C2), subsequent rows show variations; hyperparameter values are only shown if they differ from the reference.

| | Hyperparameters | | success rate |
|---|---|---|---|
| | $\varepsilon_{\mathrm{i}}$ | $T_\varepsilon$ | |
| **C2 (ref.)** | **0.3** | **5M** | **98 %** |
| | 0.6 | | 93 % |
| | | 10M | 85 % |

Table 9: Best test success rates on random test sets of $500$ equations of type (7) with $a_i, b_i \in \mathbb{Z}$, $a_0 = b_0 = a_3 = b_3 = 0$ or $a_1 = b_1 = a_2 = b_2 = 0$, and $p_0 = 0$ (cf. A.1) for hyperparameter variations of model S1 (cf. Fig. 3 of the main text and Table 3). First row is the reference (S1), subsequent rows show variations; hyperparameter values are only shown if they differ from the reference.

| | Hyperparameters | | | | | success rate |
|---|---|---|---|---|---|---|
| | hidden layers | $p$ | $\varepsilon_{\mathrm{i}}$ | $T_\varepsilon$ | $\eta$ | |
| **S1 (ref.)** | **$16\,000, 8000, 4000, 2000$** | **4** | **1** | **5M** | **0.05** | **93 %** |
| | | 8 | | | | 93 % |
| | | | | 10M | | 80 % |
| | $8000, 4000, 2000$ | | 0.6 | | | 90 % |
| | $8000, 4000, 2000$ | | 0.6 | 10M | 0.01 | 3 % |
| | $8000, 4000, 2000$ | | 0.6 | | 0.01 | 5 % |

Table 10: Best test success rates on random test sets of $500$ equations of type (7) with $a_i, b_i \in \mathbb{Z}$ and $p_0 = \frac{1}{2}$ (cf. A.1) for hyperparameter variations of model S2 (cf. Fig. 3 of the main text and Table 3). First row is the reference (S2), subsequent rows show variations; hyperparameter values are only shown if they differ from the reference.

| | Hyperparameters | | | | | success rate |
|---|---|---|---|---|---|---|
| | $p$ | $\varepsilon_{\mathrm{i}}$ | $\varepsilon_{\mathrm{f}}$ | $T_\varepsilon$ | $\eta$ | |
| **S2 (ref.)** | **4** | **0.3** | **0.1** | **10M** | **0.05** | **49 %** |
| | 8 | | | | | 37 % |
| | | 0.6 | | | | 36 % |
| | | 0.9 | | | | 5 % |
| | | 0.1 | 0.05 | | 0.01 | 34 % |
| | | | | 10M | | 6 % |

Table 11: Best test success rates on random test sets of $500$ equations of type (7) with $a_i, b_i \in \mathbb{Z}$ and $p_0 = \frac{2}{3}$ (cf. A.1) for hyperparameter variations of model S3 (cf. Fig. 3 of the main text and Table 3). First row is the reference (S3), subsequent rows show variations; hyperparameter values are only shown if they differ from the reference.

| | Hyperparameters | | | success rate |
|---|---|---|---|---|
| | $p$ | $\varepsilon_{\mathrm{i}}$ | $\eta$ | |
| **S3 (ref.)** | **8** | **0.6** | **0.01** | **78 %** |
| | 4 | | | 74 % |
| | 4 | 0.3 | | 75 % |
| | | 0.3 | 0.02 | 57 % |
| | | 0.3 | 0.05 | 71 % |
| | 4 | 0.3 | 0.02 | 75 % |
| | 4 | 0.3 | 0.05 | 66 % |

Table 12: Best test success rates on random test sets of $500$ equations of type (7) with $a_i, b_i \in \mathbb{Q}$ and $p_0 = \frac{2}{3}$ (cf. A.1) for hyperparameter variations of model S4 (cf. Fig. 3 of the main text and Table 3). First row is the reference (S4), subsequent rows show variations; hyperparameter values are only shown if they differ from the reference.

| | Hyperparameters | | | success rate |
|---|---|---|---|---|
| | $p$ | $\varepsilon_\mathrm{i}$ | $\eta$ | |
| **S4 (ref.)** | 4 | 0.2 | 0.05 | **80** % |
| | 8 | | | 77 % |
| | | | 0.01 | 75 % |
| | 8 | | 0.01 | 70 % |
| | 8 | 0.6 | 0.01 | 30 % |

Table 13: Best test success rates on random test sets of $500$ equations of type (7) with $a_i, b_i \in \mathbb{Q}$ and $p_0 = \frac{2}{3}$ (cf. A.1) for hyperparameter variations of model S5 (cf. Fig. 3 of the main text and Table 3). First row is the reference (S5), subsequent rows show variations; hyperparameter values are only shown if they differ from the reference.

| | Hyperparameters | | | | success rate |
|---|---|---|---|---|---|
| | $p$ | $\hat{\varepsilon}$ | $\varepsilon_\mathrm{i}$ | $\varepsilon_\mathrm{f}$ | |
| **S5 (ref.)** | 4 | 1 | 0.2 | 0.01 | **85** % |
| | 8 | | | | 90 % |
| | | 0.1 | | | 83 % |
| | | | 0.1 | | 81 % |
| | | | | 0.05 | 87 % |
| | 8 | | | 0.05 | 83 % |
| | | | 0.4 | 0.05 | 84 % |

Table 14: Best test success rates on the test dataset of $10\,000$ equations of type (6) with $a_i \in \mathbb{Q}$ (cf. A.1) for hyperparameter variations of model AR (cf. Fig. 4 of the main text and Tables 3 and 4). First row is the reference (AR), subsequent rows show variations; hyperparameter values are only shown if they differ from the reference.

| | generator hyperparameters | | solver hyperparameters | | | | success rate |
|---|---|---|---|---|---|---|---|
| | $\varepsilon_\mathrm{i}$ | $\eta_\mathrm{i}$ | $p$ | $\gamma$ | $\varepsilon_\mathrm{i}$ | $\eta_\mathrm{f}$ | |
| **AR (ref.)** | 0.5 | 0.01 | 8 | 0.9 | 0.5 | 0.005 | **74** % |
| | | | 4 | | | | 67 % |
| | | | | 0.95 | | | 3 % |
| | | 0.05 | | | | 0.001 | 86 % |
| | 0.8 | | | | 0.8 | | 56 % |

Table 15: Best test success rates on the test dataset of $10\,000$ equations of type (7) with $a_i, b_i \in \mathbb{Q}$ and $p_0 = \frac{2}{3}$ (cf. A.1) for hyperparameter variations of model AS1 (cf. Fig. 4 of the main text and Table 3). First row is the reference (AS1), subsequent rows show variations; hyperparameter values are only shown if they differ from the reference.

| | Hyperparameters | | success rate |
|---|---|---|---|
| | $p$ | $\gamma$ | |
| **AS1 (ref.)** | 8 | 0.9 | **16** % |
| | | 0.95 | 5 % |
| | 4 | | $< 1$ % |
| | 4 | 0.95 | $< 1$ % |

Table 16: Best test success rates on the test dataset of $10\,000$ equations of type (7) with $a_i, b_i \in \mathbb{Q}$ and $p_0 = \frac{2}{3}$ (cf. A.1) for hyperparameter variations of model AS2 (cf. Fig. 4 of the main text and Table 3). First row is the reference (AS2), subsequent rows show variations; hyperparameter values are only shown if they differ from the reference.

| | Hyperparameters | | | | | success rate |
|---|---|---|---|---|---|---|
| | $p$ | $\gamma$ | $\varepsilon_i$ | $\eta_f$ | $\alpha_\eta$ | |
| **AS2 (ref.)** | **8** | **0.95** | **0.3** | **0.005** | **0.5** | **99 %** |
| | | 0.9 | | 0.01 | | 98 % |
| | 4 | 0.9 | 0.5 | | | 59 % |
| | | 0.9 | 0.5 | | 1 | 61 % |

# B  DATASETS

The archive `sm_datasets.zip` contains the following files:

- `lin-cint-N1000.txt, lin-cint-N10000.txt`: Test datasets of 1000 and 10 000 equations with integer coefficients (main text Eq. (6), $a_i \in \mathbb{Z}$).

- `lin-crat-N1000.txt, lin-crat-N10000.txt`: Test datasets of 1000 and 10 000 equations with rational coefficients (main text Eq. (6), $a_i \in \mathbb{Q}$).

- `lin-ccpxint-N1000.txt, lin-ccpxint-N10000.txt`: Test datasets of 1000 and 10 000 equations with complex-integer coefficients (main text Eq. (6), $a_i \in \mathbb{Z} + i\mathbb{Z}$).

- `lin-ccpxrat-N1000.txt, lin-ccpxrat-N10000.txt`: Test datasets of 1000 and 10 000 equations with complex-rational coefficients (main text Eq. (6), $a_i \in \mathbb{Q} + i\mathbb{Q}$).

- `lin-cint+0.5sym1-N10000.txt`: Test dataset of 10 000 equations with symbolic and integer numerical coefficients (main text Eq. (7), $a_i, b_i \in \mathbb{Z} + i\mathbb{Z}$, $p_0 = \frac{1}{2}$, cf. Methods).

- `lin-crat+0.33sym1-N10000.txt`: Test dataset of 10 000 equations with symbolic and rational numerical coefficients (main text Eq. (7), $a_i, b_i \in \mathbb{Q} + i\mathbb{Q}$, $p_0 = \frac{2}{3}$, cf. Methods).

- `fig2a.txt, fig2b.txt, fig3.txt, fig4a.txt, fig4c.txt`: Source data of figures in the main text. Space-separated test success rates (`test_success_*` columns) and average number of steps (`avg_steps_*` columns) for all training and test configurations, cf. the headers in each file.

- `fig4b.txt`: Source data of Fig. 4b in the main text. Space-separated generator frequencies (`generated('...')`) and solver success rates (`solved('...')`) for all equation classes, cf. the file's header.

