# OpenReview forum: "Symbolic equation solving via reinforcement learning"
_ICLR.cc/2024/Conference — Submitted to ICLR 2024_

### Official Review · Reviewer_QB4J · 2023-10-30

**Soundness:** 2 fair
**Presentation:** 2 fair
**Contribution:** 2 fair
**Rating:** 5
**Confidence:** 3

**Summary:**

The paper proposes a new method for solving algebraic equations via reinforcement learning, in a manner akin to a Computer Algebra System.  However, the key contribution of the paper is that the solving strategy is learnt by a reinforcement learning (RL) system. The RL system has access to a set of rules/actions, that can be iteratively composed to solve the linear equation. The authors propose a novel strategy, whereby parts of equations and additional constant coefficients can be stored in a stack, from where they can be called or acted upon with a possible set of actions, to solve the linear equation.

**Strengths:**

- The proposed strategy, though simple is quite original
- The writing is clear
- The experimental analysis seems sound

**Weaknesses:**

- The problem is interesting but the restriction to just linear equations is quite severe
- I am not sure if this is a widely different approach from Computer Algebra System (CAS), as in the end CASs also implement a search strategy in a space of possible actions. This is not completely a criticism, as it could be exciting to extend CASs with RL for efficiency. But I am not sure this paper provides many novel ideas in that direction. I believe the paper can be a stepping stone to a real proof-of-concept for RL applications to CAS, but in its present form it is too limited.

**Questions:**

- Have you looked into expanding the existing open-source CAS (such as Sympy) systems with RL?
- Why have you not experimented with quadratic equations? does the space of actions significantly explodes in that case?
- In what case if ever, exponential (^) is used?

---

> ### Author Response · Authors · 2023-11-20
>
> We thank the reviewer for the thorough reading of our submission and the valuable feedback.
>
> Regarding the weaknesses they identified:
>
> >  -  The problem is interesting but the restriction to just linear equations is quite severe
>
> A similar remark was made by Reviewer hHWo. We agree that the problem of solving linear equations is relatively simple. Yet we do believe that the addition of symbolic coefficients, in particular, adds some complexity to the problem, and we are working on extensions to other domains. The advantage of linear equations, and one reason why we chose them, is that this is a problem that is well understood by humans and that it is relatively straightforward to follow and analyze the RL agent's solution strategy, which we deemed helpful given that this is the first exploration of our approach/idea. Furthermore, the problem complexity we can handle is ultimately limited by the computing resources available to us.
>
> We have adjusted the discussion/conclusions to work out more clearly why we think that our results are nontrivial.
>
> >  -  I am not sure if this is a widely different approach from Computer Algebra System (CAS), as in the end CASs also implement a search strategy in a space of possible actions. This is not completely a criticism, as it could be exciting to extend CASs with RL for efficiency. But I am not sure this paper provides many novel ideas in that direction. I believe the paper can be a stepping stone to a real proof-of-concept for RL applications to CAS, but in its present form it is too limited.
>
> We agree with this interpretation of the RL agent performing a search for solution strategies in the space of equation transformations. In fact, we see it as one of our main contributions that we found a way to make such problems requiring exact mathematical transformations amenable to machine-learning/reinforcement-learning methods. We rephrased parts of the conclusions to highlight this better.
>
> Regarding their questions:
>
> > - Have you looked into expanding the existing open-source CAS (such as Sympy) systems with RL?
>
> Yes, we would even argue that our current implementation already is an "expansion" of SymPy in some sense. Essentially, we replaced the "equation solving" module of SymPy with our RL agent, but the term handling is left to SymPy. This is briefly explained at the end of Sec. 2.3 with details in Appendix A.7. So our approach is already integrated with SymPy and can easily be adapted to replace or extend other SymPy functionality in principle.
>
> We have added a comment at the end of Sec. 2.3. to highlight this integration with SymPy.
>
> > - Why have you not experimented with quadratic equations? does the space of actions significantly explodes in that case?
>
> We are currently experimenting with quadratic equations as a natural next step. Quadratic equations come with additional complications like multiple solutions and rational powers (square roots) for inverse operations, which can lead to non-polynomial forms of equations during the RL agent's explorations, for example. We do not include results in this paper because we are still investigating the best strategy and are not satisfied with the success rates yet.
>
> > - In what case if ever, exponential (^) is used?
>
> In the present context of linear equations, the exponential (^) operation is used/required only to invert products. For example, to solve the equation 2\*x = 4, the agent will want to multiply both sides by 1/2, and the typical steps to prepare this transformation will be as follows: push the coefficient in front of x (here 2) to the stack, push the constant -1 to the stack, apply ^ on the stack (resulting in 2^(-1) = 1/2 as the top element of the stack), apply \* on the equation. This is also exemplified in Fig. 2d, see the second-to-last action in particular. Of course, the agent explores other uses of ^ during training, but usually learns quickly that these are not helpful for the problems at hand.

---

### Official Review · Reviewer_hHWo · 2023-10-31

**Soundness:** 2 fair
**Presentation:** 4 excellent
**Contribution:** 1 poor
**Rating:** 5
**Confidence:** 5

**Summary:**

The authors introduce a simple computer algebra system for manipulating elementary algebraic equations. This system comprises a sequence-based representation for the algebraic equations, and a small stack based machine for executing manipulations of these expressions. The authors then train a deep reinforcement learning agent, using double deep Q-learning, to operate this stack machine, with the objective of reducing given algebraic expressions to a canonical ("solved") form.

The authors show that, with an appropriate curriculum, the agent can indeed learn to operate this machine to achieve the goal of solving the equations. Further, they show that by introducing a second RL agent in an adversarial generator-solver arrangement with the first, that they can reduce the need to hand-craft a curriculum - although they do find that it is still useful to craft a simpler curriculum for maximum performance.

**Strengths:**

The manuscript is very clearly written. I found it very easy to read and understand what the authors had done. The presentation of the results was direct, easy to understand, and helpful.

**Weaknesses:**

I think there are two main weaknesses with this paper: that it doesn't support its main claim; and that the domain it is applied to is too simple to really get a sense for whether the approach is useful or interesting.

The paper claims in the introduction that "humans must implement the discovered rules as computer programs" in traditional computer algebra systems and that "this process could benefit greatly from techniques that enable computers to discover and implement transformation rules on their own." In the conclusion the authors claim that their work "can be seen as a first step towards the general goal of creating a machine-learned computer algebra system in which the fundamental laws of mathematical reasoning and deduction are discovered autonomously by an AI." I do not think this claim is supported by the work presented. When the authors introduce their representation of the algebraic equations, and the operations of the stack machine, they implicitly encode all of the "fundamental laws of mathematical reasoning and deduction" that are necessary for this domain. These are fully sufficient for this domain, and so in that sense also all the "fundamental laws" that this system will ever contain. More specifically, their assumptions immediately partition the space of expressions into equivalence classes that encode the notion of semantic equality in this domain. The act of "solving" the equations can be thought of as the act of finding a canonical exemplar (or, at least, an exemplar from a canonical subset as the authors' definition of "solved" admits multiple solutions). So what the RL algorithm has found is a search algorithm within an equivalence class that finds a canonical example. This is an interesting and useful thing to do, but I would argue that is does not in any sense enable their system to discover any fundamental laws of mathematics - these were all in there right from the start when the authors defined their system. So I don't think the main claim of the paper is supported by the developments presented.

The second weakness is that the domain in which the authors work is exceedingly simple: that of simple algebraic equalities. Viewed through the lens described above - that what the authors' RL algorithm really does is discover effective search procedures for canonical exemplars in the domain, then I think a valid question is "how complex would it be to develop such an algorithm another way?" And the answer is "essentially trivial". Many straightforward algorithms exist for doing this search, including the ones routinely taught to schoolchildren and those implemented in standard linear system solvers. So from my perspective showing that an RL agent can discover such an algorithm isn't really a convincing result. It would be interesting if an RL algorithm could find search algorithms that are not known to existing computer algebra systems (or even schoolchildren) but that hasn't been demonstrated in this paper. So my feeling here is that the authors would really have to show that their system can discover non-trivial search algorithms for it to be a notable result.

**Questions:**

The opening analogy is confusing: while there's only one correct solution to the equation in some sense, there are usually multiple structural forms for that solution (x = 2 - c, x = -c + 2), and there are many sequences of manipulations that would lead to those goal states. So it seems like the situation is not that different from chess, in the sense that one is trying to find a sequence of moves to get to a subset of states that have some particular property.

---

> ### Author Response · Authors · 2023-11-20
>
> We thank the reviewer for the thorough reading of our submission and the valuable feedback.
>
> Regarding the weaknesses they identified:
>
> > The paper claims in the introduction ...  I don't think the main claim of the paper is supported by the developments presented.
>
> While we understand the reviewer's point here, we disagree with their interpretation that our RL agent does not discover rules of mathematical reasoning and deduction. The reviewer states that we "implicity encode all of the 'fundamental laws of mathematical reasoning and deduction' that are necessary for this domain". We agree with this assessment, but we continue to believe that our RL agent still adds new rules or mathematical insights to this foundation: It learns how to adopt the laws it has been equipped with to carry out a task that it had not known about before. We would argue that this is, in fact, the standard procedure in mathematical research: Starting from some definitions and previously established results, the mathematician combines them in an expedient way to arrive at something that was not obvious at the outset. So in some sense, all mathematical reasoning is just a "search algorithm" in the action space of chaining together axioms and previous insights. We see it as one of our main contributions that we found a way to make such a "theory space" of exact mathematical transformations amenable to machine-learning/reinforcement-learning methods.
>
> We have rephrased and expanded the corresponding text passages in the conclusions, acknowledging in particular that the previous wording speaking of "fundamental laws" may be perceived as somewhat pretentious.
>
> > The second weakness is ...  the authors would really have to show that their system can discover non-trivial search algorithms for it to be a notable result.
>
> A similar remark was made by Reviewer QB4J. We agree that the problem of solving linear equations is relatively simple. Yet we do believe that the addition of symbolic coefficients, in particular, adds some complexity to the problem, and we are working on extensions to other domains. The advantage of linear equations, and one reason why we chose them, is that this is a problem that is well understood by humans and that it is relatively straightforward to follow and analyze the RL agent's solution strategy, which we deemed helpful given that this is the first exploration of our approach/idea. Furthermore, the problem complexity we can handle is ultimately limited by the computing resources available to us.
>
> The reviewer mentions schoolchildren, and we think that they are actually a great example to illustrate why learning to solve algebraic equations (a) involves new rules of mathematical reasoning and deduction ("first weakness") and (b) may not be so trivial after all ("second weakness"). When humans learn how to solve linear equations in middle school, they build on an extensive basic mathematics education from previous years, yet it is undeniable that once they mastered how to deal with equations, they have acquired a new skill. Our RL agent is similar to such a middle-school student, it already knows how to manipulate mathematical terms, but now it discovers (on its own) how to use this knowledge to solve equations. Furthermore, it may be "essentially trivial" to develop a solver _if one already knows how to solve equations_. However, we suspect that asking the average middle-school student to devise such a solver _before_ they are taught the canonical strategy would most likely not be very effective ...
>
> We have adjusted the discussion/conclusions to work out more clearly why we think that our results are nontrivial.
>
> Regarding their questions:
>
> > The opening analogy is confusing: while there's only one correct solution to the equation in some sense, there are usually multiple structural forms for that solution (x = 2 - c, x = -c + 2), and there are many sequences of manipulations that would lead to those goal states. So it seems like the situation is not that different from chess, in the sense that one is trying to find a sequence of moves to get to a subset of states that have some particular property.
>
> We can see the reviewer's point. The idea of this analogy was that equations have a unique solution up to certain "symmetry transformations" (e.g., commutativity of addition in the example). In chess, on the contrary, there are many possible outcomes of a game that are not equivalent by any meaningful symmetry. But we understand that the analogy can be confusing and may not get our idea across, hence we have removed it from the introductory paragraph.

---

> > ### Comment · Reviewer_hHWo · 2023-12-05
> >
> > [Administrative note: I was unable to add this comment addressed to the authors during the discussion period, as it ended before I was able to read the authors' responses. In the absence of a reply from the AC on how to rectify this, I am going to leave my reply to the authors as a comment here.]
> >
> > I thank the authors for their response, as well as their response to the other reviewers questions, all of which I have read.
> >
> > I think the authors' response to my first comment is reasonable, and I note that their proposed rewording of the paper does address my criticism in the main. If I may expand on the comment in my review, in the context of the authors' reply. It feels like there are two separate things that could be discussed here: where the boundary is between simply pushing symbols around and actually generating mathematical insights; and whether all mathematical discovery can be thought of as symbol pushing. On the former point, I think the authors are right to note that symbol pushing goes a very long way, and there are certainly many things that can be learned from application of the existing rules that could be counted as genuine mathematical developments. I think that perhaps the very simple example domain used in the paper rather disguises this aspect of the work, and I think that the authors' clarification plus rewording does a good job of highlighting that this is an initial step towards work that could be used to generate mathematical insight. So in this sense, I think the authors have addressed my criticism.
> >
> > I do think, though, that a lot of impactful, fundamental mathematical work does not fall into the category of working within the existing rules ... rather it is better thought of as reformulating the system of rules. I do not think a system like the authors' could ever produce this sort of result, so I do think it is wise not claim too much about "fundamental laws" of mathematics. I am glad to see the authors have softened this claim.
> >
> > In terms of my second criticism, that the example mathematical system - linear equations - that the authors have used to demonstrate their system is too simple: I think this criticism, from my perspective, still stands. It's certainly a neat result that an RL algorithm can learn to use a stack-based calculator to do simple mathematical rearrangements. But I think to be of significant note, and thus suitable or publication in ICLR, I'd want to see a result that gave me confidence that further work on this area might lead to more impressive, far-reaching results. I think the simplicity of the authors' chosen problem domain does not allow the work to demonstrate that convincingly.
> >
> > So, overall, it still feels to me that what the authors have really shown is an interesting approach to using RL to generate new search techniques in computer algebra. And I think for a paper on that to be really notable, it would have to show that it can compete with the existing techniques for executing such searches, on a non-trivial problem domain.
> >
> > I have adjusted my score to reflect the authors' revised positioning of the work, but I still think it does not meet the bar for acceptance.

---

### Official Review · Reviewer_8V3F · 2023-11-02

**Soundness:** 2 fair
**Presentation:** 3 good
**Contribution:** 2 fair
**Rating:** 3
**Confidence:** 2

**Summary:**

The authors present a reinforcement learning approach for solving linear equations and evaluate it on a set of problems.

**Strengths:**

- the problem domain is interesting, there is not much work on using RL to solve symbolic equations
- The paper in generall is easy to understand
- The related work appears to be covered

**Weaknesses:**

There is no clear motivation why the proposed approach is a good idea.
The authors state that:
" Evidently, this process could benefit greatly from techniques that enable computers to
discover and implement transformation rules on their own. Moreover, finding viable approaches
to do so will eventually help to make machine-learning models more adept at mathematics and
problems requiring exact solutions in general."

But why is it a disadvantage that current automatic equation solvers integrate human expert knowledge? What is the pain point of the current solutions that exist?

2. There are no comparisons done in this paper. This is striking both on the small as as well as the large scale.

2.1 how much faster/slower and more/less accurate does the proposed method work compared to established methods from Mathematica, Maple, Matlab, or SymPy?

2.2  What are the impacts of the hyper-parameters on the performance of the solution? E.g. how relevant is the discount factor, the complexity of the network? While I understand that deep RL approahces have a lot of hyper-parameters it is still relevant to identify the most sensitive ones  and do some form of inspection and analysis regarding the robustness of the approach.

2.3  What are the impacts of the RL approach (double Q learning) on the solutions? How well would  for instance PPO methods compare?

2.4 Compared to related work, is there no related method you could compare to?

**Questions:**

After training, what is the sucess rate compared to established solvers?
What is the wall-clock time in inference compared to established solvers?

---

> ### Author Response · Authors · 2023-11-20
>
> We thank the reviewer for the thorough reading of our submission and the valuable feedback.
>
> As a general remark, we would like to emphasize that the goal of our study was not to come up with an RL framework that outperforms established computer algebra systems (CAS) like Mathematica, Maple, Matlab, or Sympy. If this were the case, the domain of linear equations would be too simplistic as all those existing CAS will solve such equations with perfect success rates and almost instantly for all practical purposes. Instead, our goal was to devise a viable strategy that enables an RL agent to discover transformation rules typical for CAS operation on its own, without human intervention. Admittedly, our present result does not give a practical advantage for solving linear equations. Nevertheless, it showcases a way towards building machine-learning models for exact mathematics, a domain that has proved to be exceptionally hard to master. In the long run, we hope that such a strategy can be used to aid mathematical research and discover relations that have not been known to humans before.
>
> Regarding the weaknesses the reviewer identified:
>
> > There is no clear motivation why the proposed approach is a good idea. (...) But why is it a disadvantage that current automatic equation solvers integrate human expert knowledge? What is the pain point of the current solutions that exist?
>
> The pain point, in our opinion, is that humans are relatively slow at discovering and implementing mathematical relations and rules. Current CAS build on thousands of years of mathematical research and the major ones have been under development for over 30 years to "teach" computers what humans know. Of course, we are well aware that, presently, our approach cannot compete with these established software packages and does not discover anything unknown to humans. However, we still believe that it is a first step towards automated ways of mathematical research and that such an autonomous "RL researcher" could greatly enhance the power of CAS as well as the human mathematical knowledge base. We emphasize that it will still build on established human insights (like our present approach does, too), but it should be capable of exploring new domains that it had not been explicitly taught about in an independent, autonomous way.
>
> We have adjusted the introduction to explain more clearly in what sense we aim to enable computers to discover and implement transformation rules on their own.
>
> > 2.1 how much faster/slower and more/less accurate does the proposed method work compared to established methods from Mathematica, Maple, Matlab, or SymPy?
>
> We did not carry out such a comparison because this is not the aim of our study as outlined above. All of the established CAS will achieve success rates of essentially 100 % for the problem of linear equation solving. They will most probably run faster than our method, too, which has not been optimized for speed yet. It will not matter for the human user who is interested in solving a particular equation because they won't experience a noticeable delay, but it might become relevant if one wishes to solve a batch of several hundreds of equations at a time.
>
> > 2.2 What are the impacts of the hyper-parameters on the performance of the solution? E.g. how relevant is the discount factor, the complexity of the network? While I understand that deep RL approahces have a lot of hyper-parameters it is still relevant to identify the most sensitive ones and do some form of inspection and analysis regarding the robustness of the approach.
>
> A qualitative overview of the influence of the various hyperparameters can be found in Appendix A.8. As explained there, we did not carry out a systematic grid search due to the large number of hyperparameters, but we explored several different variations for each type of problem. In the revised manuscript, we have added Tables 5-16, which list the success rates for hyperparameter variations of each of the models from Figs. 2-4 to provide a quantitative account of the hyperparameter influences.
>
> (continuing below)

---

> > ### Author Response · Authors · 2023-11-20
> >
> > (continued reply)
> >
> > > 2.3 What are the impacts of the RL approach (double Q learning) on the solutions? How well would for instance PPO methods compare?
> >
> > At present, we have not explored other RL approaches on this type of problem. Exploring different RL approaches like PPO is certainly an interesting suggestion and may become helpful when extending to other problem domains. Given the discrete nature of the action space, we did not see any obvious reasons why other methods should be preferred in our present setting, and evidently Q learning worked. We _did_ experiment with different variants of Q learning in the beginning (e.g., single vs. double Q learning; experience prioritization; loss function amendments like convergent DQN or weight decay; multistep action lookahead). Compared to the implementation detailed in Appendix A, those alternatives did not yield any significant advantages, but required longer training times and sometimes performed worse.
> >
> > > 2.4 Compared to related work, is there no related method you could compare to?
> >
> > The most closely related works that we are aware of have been discussed in Sec. 4.1, but none of them address the exact same problem, so we don't see a reasonable way to compare them quantitatively to our results.
> >
> > Regarding their questions:
> >
> > > After training, what is the sucess rate compared to established solvers? What is the wall-clock time in inference compared to established solvers?
> >
> > See reply to item 2.1 above.

---

### Meta-Review · Area_Chair_VT2J · 2023-12-15

**Metareview:**

This paper proposes using reinforcement learning to learn transformation rules for solving linear algebra problems.  The reviewers had various issues with the paper.  The main problem seems to be that the domain is quite simple and algorithmic solutions are known.  There is considerable interest in whether LLMs can do mathematics but this paper is not about LLMs.  More generally there is no clear metric for evaluating this paper and hence the contribution is unclear.

**Justification For Why Not Higher Score:**

The significance of this work is not established in the paper.

**Justification For Why Not Lower Score:**

This is the lowest score.

---

### Decision · Program_Chairs · 2024-01-16

Reject